# Protections in the Recreational Practice of Ski and Snowboard—An Age and Gender Discussion? A Case Study in Spain

**DOI:** 10.3390/jfmk6040084

**Published:** 2021-10-14

**Authors:** Marcos Mecías-Calvo, Carlos Lago-Fuentes, Iker Muñoz-Pérez, Jon Mikel Picabea-Arburu, Rubén Navarro-Patón

**Affiliations:** 1Facultad de Formación del Profesorado, Universidade de Santiago de Compostela, 27001 Lugo, Spain; marcos.mecias@usc.es (M.M.-C.); ruben.navarro.paton@usc.es (R.N.-P.); 2Facultad de Ciencias de la Salud, Universidad Europea del Atlántico, 39011 Santander, Spain; carlos.lago@uneatlantico.es (C.L.-F.); jon.picabea@uneatlantico.es (J.M.P.-A.); 3Sport Training, RUNNEA, 48901 Barakaldo, Spain; 4Facultad de Ciencias de la Salud, Universidad Isabel I, 09003 Burgos, Spain; 5Physical Education and Sport Department, Faculty of Education and Sport, University of the Basque Country UPV/EHU, 01007 Vitoria-Gasteiz, Spain

**Keywords:** sports protections, winter sports, recreational practice

## Abstract

The objective of this research was to evaluate the protections taken by ski and snowboard recreational athletes of the winter stations Sierra Nevada and Alto Campoo (Spain), regarding gender, age, and practiced sport. A total of 520 users participated, Sierra Nevada (*n* = 306 (58.8%)); Alto Campoo (*n* = 214 (42.2%)), 257 of them were men (49.4%) and 263 (50.6%) were women, from 6 to 64 years old, classified by 4 stages of development (Childhood (*n* = 106 (20.4%)); Teenagers (*n* = 110 (21.2%)); Youth (*n* = 101 (19.4%)); Adults (*n* = 203 (39.0%))). For the data collection, an Ad Hoc questionnaire was used (Socio-demographic data, use/no use of protection). The data revealed that 23.5% of the participants did not use any protection. Regarding the development stage, 1% of the children did not use any protection, neither did 3.1% of the teenagers, 6.7% of the youth, or 12.7% of the adults (*p* < 0.001). Regarding gender, a total of 17.1% of men did not use protection, and regarding women, 6.3% of them did not use it (*p* < 0.001). In relation to the practiced sport, 15.8% of the skiers did not use it against 7.7% of the snowboarders (*p* = 0.006). The use of protection for the practice in winter sports is not enough to reduce the injury risk in these sports and, in the worst cases, fatal accidents.

## 1. Introduction

Winter sports have had a significant development regarding the number of participants during the last 20 years [1], and ski or snowboard are two of the most popular in the world [2]. The winter stations have approximately 400 million visits around the world [3] so an increase in the registered injuries associated with these winter sports can be expected.

Winter sports practitioners should always have in mind the main factors related to the chance to suffer an accident and an injury [1,4]. The internal ones are those related to physiological and psychological variables, such as muscular strength or anxiety. The external factors include the snow state, the weather conditions, the slope, or the used equipment, among others [5]. These factors are related to a substantial risk of sports injuries, with a reported incidence of 0.5–1.35 injuries per 1000 skier/snowboarder days on the recreational ski and snowboarding practice [6,7,8]. Among others, the cause of traumatic injuries is related to the use of the proper protections [4,9], which could be avoided or decreased if the correct equipment was used [10,11,12,13,14]. These kinds of injuries are more commonly suffered in adult men [15,16] who practice these sports. In addition, the severity of these injuries is higher for those who do not use helmets [12,17]. Regarding young skiers or snowboarders, the use of helmets does not increase the incidence of spine injuries and reduces the incidence of heavy head injuries, which need treatment [18]. Hackam, Kreller, and Pearl [19] affirm that “snow sports injuries, particularly snowboarding, cause severe childhood morbidity. Helmet usage, training requirements, and regulation of licensed resorts may reduce the morbidity and staggering costs”. Provance, Engelman, and Carry [20] tested that the behavior of parents using their helmets is strongly associated with the use of the helmet by the child or teenager. In this line, the use of helmets does not imply risky behavior [21].

Usually, the use of protective equipment can explain the decrease of upper and lower members and head injuries during the last few years [9]. The use of the helmet for the practice of winter sports can reduce the risk of head injuries from 15 to 60% [16,22,23]. Notwithstanding, more than 25% of skiers and snowboarders do not use it [24].

Another popular protection for the winter sports practice are wrist guards, which show their efficiency in the protection against snowboarders’ wrist injuries [25,26]. Regarding this, a recent study showed that children are two times more likely to suffer wrist injuries than adults when snowboarding [27], which highlights the need to know in-depth the frequency of use and reasons for these protections regarding age.

As has been shown, previous studies have analyzed the incidence and burden of injury in other countries [28]; in elite winter athletes; and the relationship among protections and some type of injury, especially in the lower limbs [29]. Regarding the case of Spain, previous reports have shown that more than 5 million people visit every year ski resorts to practice ski and snowboard, distributed in 28 winter stations along the country (https://www.atudem.es/, accessed on 21 March 2021). Regarding this, sociodemographic factors clearly influence the sports culture and the use of protections [30]. Furthermore, no previous studies related to these factors have been applied to the Spanish population. Attending this, studies that analyze the use of protections and their influencing factors in Spain are needed to properly design specific preventative strategies related to the type and frequency of use of protections. For these reasons, the objective of this study was to analyze the protections taken by recreational ski and snowboard athletes, regarding gender and age in the winter resorts of the Sierra Nevada and Alto Campoo (Spain). In this sense, this is the first study, which analyzes the frequency and type of protection used in a great sample of participants comparing the type of winter sports, gender, and development stage. This study can help to adopt new preventative strategies regarding the type of sport, define different messages to increase the frequency of use of protections according to their development stage and, also, to the gender.

## 2. Materials and Methods

### 2.1. Study Design

To answer the planned objective, a cross-sectional study design was carried out [31].

The privacy of the data given by the subjects was respected and they were also previously informed of the aim, recording techniques, and data analysis through informed consents directed to adults and underage. The Investigational Review Committee of the Department of Physical Education and Sport Sciences of a Spanish university approved the research. All research was conducted in accordance with the Declaration of Helsinki and meets the European General Data Protection Regulation (EU GDPR; EU 2016/679).

### 2.2. Participants

The data selection was non-probabilistic regarding the subjects from both winter stations (Sierra Nevada and Alto Campoo), which freely decided to participate in the study by answering a questionnaire [32]. Among other inclusion criteria, the subjects must have had more than 6 years old and less than 64, to categorize them regarding the development stages in which the sample was divided: childhood (6 to 11 years old); teenagers (12 to 17 years old); young adults (18 to 24 years old) and adults (25 to 64 years old) [33]. As exclusion criteria, all the subjects participating in a high-level competition were not taken into account.

### 2.3. Procedures

An ad-hoc questionnaire was designed by a panel of experts, composed of experts in qualitative methods (questionnaires) and sports scientists specialized in injury prevention. It was composed of two main sections: the first related to socio-demographic data and, the second one, related to the use of protection during recreational practice. After designing the questionnaire and receiving ethical approval, a pilot study of the survey was performed in Alto Campoo winter station during November 2017 (*n* = 20; childhood *n* = 4, teenagers *n* = 4, young adults *n* = 5, adults = 7) to evaluate the clarity and content of the questionnaire. After the pilot test, no changes were required on the questions designed by the panel of experts. Then, the questionnaire was provided via a printed sample in person, before skiing, between December 2017 and February 2018 in both winter stations. When participants were young adults, the questionnaire was provided to their parents to ensure their understanding of the questions and the reliability of their answers.

Participants who consented to participate in this study answered nine questions split into the following sections: socio-demographic data (age and gender), information about their level of practice in this type of sports (ski or snowboard), their type of practice (recreational vs. competitive) and the type and frequency of protections used during sports practice.

### 2.4. Statistical Analysis

To analyze the data, the variables were expressed through frequency tables. First, the data were found to follow a normal distribution using the Kolmogorov–Smirnov test. To study the association between variables the χ^2^ Pearson was used along with the Phi coefficient to compare the use of protection regarding the established variables (men vs. women) and practiced sport (ski-snowboard). Associations between gender (men vs. women) in the use or not of protection are presented as odds ratios (OR) with 95% confidence intervals (CI). To study the interaction between the stage of development (childhood, teenagers, young adults, and adults) and the use or not of protective equipment, Pearson χ^2^ was used along with the contingency coefficient. Chi-squared test was set between development stage and use/no use of protective equipment and gender. Subsequent correspondence analysis was used to evaluate the distance between the development stage and the use, or not, of protections among gender. The 24.0 version of the Statistical Package for the Social Sciences ^®®^ (SPSS) was used to analyze data, with a level of significance *p* < 0.05.

## 3. Results

A total of 520 recreational practitioners from the Winter resorts participated in this research, Sierra Nevada (*n* = 306 (58.8%)) and Alto Campoo (*n* = 214 (42.2%)), 257 of them were men (49.4%) and 263 (50.6%) women between 6 and 64 years old, classified according to 4 development stages (Childhood (*n* = 106 (20.4%); Teenagers (*n* = 110 (21.2%); Young adults (*n* = 101 (19.4%); Adults (*n* = 203 (39.0%) (Table 1).

The use or not of the protection regarding gender, the practiced sport, and the development stage are shown in Table 2. It is highlighted that 23.5% claim not to use protection while practicing ski or snowboard. Regarding gender, women claim to use more protection than men along all different development stages. Concerning the type of sport, 79% of skiers use them against 64% of snowboarders, and there are no big differences according to the development stage, except during childhood where the skiers are more than the snowboarders.

There is a statistical significative association between gender and the use or not of protection, and also a directly proportional relation (χ^2^ (1) = 35.299, *p* < 0.01; Phi = 0.261, *p* < 0.01), and, regarding this, women use more protection than men. That is, analyzing the risk (Odds ratio), women use protections 3.69 (CI 95% = 2.36–5.77, *p* < 0.001) more times than men. Regarding the practiced sport, there is also a statistical significative association among the sport (ski and snowboard) and the use or not of protection, and so a significative and inversely proportional relation (χ^2^ (1) = 7.362, *p* < 0.01; Phi = −0.119, *p* < 0.01), skiers have a greater tendency to use protection if they are compared with snowboarders (OR = 1.85; CI 95% = 1.18–2.90, *p* < 0.01). Concerning the age, it is observed an association and directly proportional relation between age and the use of protection (χ^2^ (3) = 41.916, *p* < 0.01; contingency coefficient = 0.273, *p* < 0.01), being the young adults and adults (in both sexes) who registered less use of protections.

Analyzing regarding men’s ages about the use of protection, it is observed an association and a directly proportional relation statistically significant between them (χ^2^ (3) = 35.661, *p* < 0.01; contingency coefficient = 0.373, *p* < 0.01), with the young adults’ group with lower use of protection, followed by adults. The chi-squared test showed a significant relationship between development stages and the use/no use of protective equipment in men (χ^2^ (3) = 35.661, *p* < 0.001). In women, there is also an association and a directly proportional relation between them and the use or not of protection (χ^2^ (3) = 11.384, *p* = 0.01; contingency coefficient = 0.204, *p* = 0.01), being the adult the one who presents less use of protection. Chi-squared test revealed significant relationship between development stages and the use/no use of protective equipment in women (χ^2^ (3) = 11.384, *p* < 0.01).

According to the development stages, there are several associations between the cohort and the use of some protective equipment. In this sense, the Childhood group will be the one who uses more protective supplies, followed by the Teenagers group (Table 3).

Regarding the practiced sports for men and women, there are no statistically significant differences (*p* = 0.06 and *p* = 0.56, respectively), so the use of protection does not depend on this variable. The most used protections can be observed in Table 4 and Table 5 regarding gender and practiced sport (Table 4) and range of age (Table 5).

As it can be observed, during ski practices, only just helmets are used, while snowboarders use also spinal, wrist, and knee protectors and padded shorts.

Table 4 represents the percentage of types of protection regarding gender and practiced sport. It is observed that both men and women skiers only use their helmet. Chi-squared test showed significant relationship between sex and the use of helmet in skiers (χ^2^ (1) = 20.9, *p* < 0.001) and snowboarders (χ^2^ (1) = 9.97, *p* = 0.002). Besides, women are prone to use this protective equipment more than men in ski (OR = 3.18; CI 95% = 1.91–5.29) and in snowboard (OR = 4.95; CI 95% = 1.73–14.2). Regarding men and the use of helmet, they do not tend to use it as much as women do in ski (OR = 0.315; CI 95% = 0.19–0.52) and snowboard (OR = 0.20; CI 95% = 0.07–0.58).

Table 5 shows the percentage of types of used protection regarding gender and range of age. All the stages generally use helmets. During youth ages (from childhood to young adults), this is the only material used, meanwhile, adults wear other types of protection such as spinal protectors, knee protectors, wrist protectors, and padded shorts.

## 4. Discussion

The objective of this study was to analyze the protections taken by recreational ski and snowboard practitioners, regarding gender and age in the winter resorts of Sierra Nevada and Alto Campoo (Spain).

Despite the benefit of using protections while practicing ski or snowboard, to decrease injuries [8], or minimizing the consequences [9], only 76.5% of the participants claimed to use them (Table 1). This percentage is slightly higher than the one reached by Martins et al. [1] who showed that 61.1% of Portuguese skiers and snowboarders used some protection during their practice. Furthermore, our results are in concordance with data reported by Ekeland and colleagues [27] in Norway, where 72% and 62% of injured skiers and boarders used helmets as protective equipment. However, these data only showed the use of injured ones, which makes it difficult to compare the overall protective equipment used with our results.

Regarding gender, it was observed that women are more prone to use protection than men (87.5% vs. 65.4%). These findings might be due to women are usually more afraid than men towards the sports practice [34,35,36], which produces a decrease in the self-efficacy of the activity, and consequently, reduces its control [37].

Comparing both sports regarding the use of protection, skiers are the ones who use it the most [38]. Martins et al. [1] did not find many differences among both sports regarding the use of protection (56.3% of the skiers and 54.9% of snowboarders). Our findings differ from Martins et al. [1], showing that 79.35% of skiers and 67.48% of boarders used some protective equipment (Table 2). Furthermore, skiers were more inclined to use some protection than snowboarders (1.85 OR). However, Martins et al. [1] included 35% of participants with advanced experience, which clearly can influence the frequency of use of protections. For this reason, our results can be inferred to the general population with a recreational level of practice in winter sports. Besides, if we take into account gender, there are no differences between women and men, because both use protection regardless of the type of sport (ski or snowboard).

According to the age, there were big differences comparing young adults and adults (from 18 to 24 and from 25 to 64 years old) and childhood and teenagers (from 6 to 11 and from 12 to 17 years old). Thus, young adults and adults use less protection, which concurs with previous studies [39,40,41]. Moreover, as our results suggest (Table 3), the childhood group displays a greater tendency of using some protective equipment than the other three groups, especially rather than young adults (OR = 10.7) and adults’ groups (OR = 9.73). This can be explained because the young adults and adults consider they have more experience, and so, more able to practice the sport, which generates a false trust and, consequently, increases the injury risk [42,43]. In this sense, Ekeland et al. [27] showed more serious injuries suffered by adults than those suffered by children, i.e., the prevalence of shoulder injuries was twice in adults. This false trust could explain partially the differences in the high number of injuries based on the age of participants.

When gender is included in the analysis, it was observed that young adult men (from 18 to 24 years old) (followed by adult men) and adult women (>24 years old) were the groups who used less protections. Furthermore, young adult men and adults were the groups who used less protection, 52.9% and 46.9%, respectively. In this sense, our results agree with previous studies, which highlighted the tendency, of young adult and adult men, to increase injury risk because of the lack of using any protective equipment [39,41] even though they are more likely to have an injury regardless of the sport [40]. This fact could explain partially the increase of injury risk in young adults and adults than in other groups [39,44,45]. This may be due to the fact that men generally assume more risks than women while practicing ski or snowboard [39,41,46]. At the same development stage, female practitioners use with more frequency any type of protection. Related to this, Giuliano et al. [47], established that men are given more attention than women when practicing sports because of cultural stereotypes, which generates a bigger difference among genders. In this sense, women could use protection to decrease their fear, and, consequently, their performance may get better. For men, the performance predictor is self-efficiency, and when it grows, the performance gets better [37].

Regarding the main reason to increase the use of protections, from the young adults to adult years, Ruedl and colleagues [48] pointed out the need to reduce the head injuries in all stage ages. However, our results showed that this hypothetical trend was not displayed. In this sense, protective equipment was less used by young adults and adult groups, stages where more injuries are described in previous studies [49].

The most commonly used protections by ski and snowboard are helmets [1]. This protective equipment has undergone a significant increase in use in the last years, more than 80% of skiers reported wearing it [16]. These are, most of the time, the only protection material skiers use during all the stages studied, while snowboarders are used to wearing also other protection materials (spinal protectors, knee protectors, wrist protectors, padded shorts), mostly from teenage years, which provides a variety of materials. Even though snowboarders, apparently, used more protective equipment, there are no differences between the rates of back injury with skiers to explain this tendency [50]. Thus, this data may be because during teenage years the athletes decide to practice snowboard more than ski, which justifies the increase of its practice in the last few years [1].

The use of helmets has increased during the last few years [1,39,51,52]. In the present study, the most used protective equipment was the helmet by far (Table 3 and Table 4), regardless of gender and age group. However, due to risky behaviors or the overestimation of the helmet protection [5,16,39], the injury ratio does not decrease [29]. On the other hand, traumatic (fall, collision) and non-traumatic death ratios have slightly decreased during the last ten years, even though traumatic deaths generated while practicing ski or snowboard are one of the most common deaths of athletes [53], which explains that ski and snowboard are sports with a high potential of traumatic injury and morbidity [54].

Finally, comparing genders regarding the use of helmets in ski and snowboard athletes, our study shows that women used them more than men (Table 4) (OR = 3.18 and OR = 4.95, for ski and snowboard respectively). As we previously explained, it may be because of their higher sense of fear, which is decreased by the use of helmets and so their performance improves. Regarding snowboarders, men generally declared no use of any protection while women claimed they do not use wrist protectors or knee protectors, which were used by men. This lack of use of protection could be related to other researches, which claimed that snowboard is riskier than ski [1,41,51] and also that injuries keep increasing, while ski injuries remain steadier [12,55].

Lastly, some limitations need to be addressed. The participants were not asked about their ability level, being it one of the factors to decide the use or not of protection [56]. However, high-level athletes were excluded, which implies that the range of ability of the sample is not so broad. The cross-sectional design of our study also cannot evaluate the evolution of the use of protections during the whole winter season neither different years. For this reason, longitudinal studies are needed in this field to compare the use of protections and apply some strategies to increase its use. Our results ought to be understood with caution since the sample in the present study is based only on two Spanish winter stations in one season. However, our results can help to design some preventative strategies in ski Spanish resorts to increase the use of protections for both ski and snowboard sports, especially for young adults and adult practitioners.

## 5. Conclusions

Despite the benefits of the protection during the practice of ski or snowboard, only 76.5% of the practitioners affirm to use them always. From this percentage, women use more protection than men, and comparing sports, the ski is the most protected sport. Regarding the stage of development, the young adults and the adults (from 18 to 24 and from 25 to 64) are the ones using less protection, both in male and female practitioners. 

Helmets are the most used protection for skiers and snowboarders (especially by women). Regarding snowboarding, there are also other protecting materials such as spinal protectors, knee protectors, wrist protectors (these last two the less used by women), padded shorts, especially from teenage years, which leads to heterogeneity of protection, even if at an adult stage some answers present the use of no protection at all.

To sum up, this information should be of interest to design adequate strategies to promote the use of protections in populations with less traditional use, such as the youngest men and, especially, when snowboarding.

## Figures and Tables

**Table 1 jfmk-06-00084-t001:** Sample’s characterization.

Item		Frequency (%)
**Gender**	Men Women	257 (49.4%) 263 (50.6%)
**Development Stage**	Childhood Teenagers Young adults Adults	106 (20.4%) 110 (21.2%) 101 (19.4%) 203 (39.9%)
**Practiced Sport**	Ski Snowboard	397 (76.3%) 123 (23.7%)
**Use of protection**	Yes No	398 (76.5%) 122 (23.5%)

Data are presented as absolute and relative frequencies.

**Table 2 jfmk-06-00084-t002:** Percentage of use of protection regarding development stage and gender.

Development Stage	Total		Men		Women		Ski		Snowboard	
	Use	No Use	Use	No Use	Use	No Use	Use	No Use	Use	No Use
**Childhood**	101 (95.3%)	5 (4.7%)	51 (92.7%)	4 (7.3%)	50 (98.1%)	1 (1.9%)	91 (94.8%)	5 (5.2%)	10 (100%)	0 (0.0%)
**Teenagers**	94 (85.5%)	16 (14.5%)	41 (77.4%)	12 (22.6%)	53 (93.0%)	4 (7.0%)	75 (83.3%)	15 (16.7%)	19 (95.0%)	1 (5.0%)
**Young adults**	66 (65.3%)	35 (34.7%)	24 (47.1%)	27 (52.9%)	42 (84.0%)	8 (16.0%)	42 (71.2%)	17 (28.8%)	24 (57.1%)	18 (42.9%)
**Adults**	137 (67.5%)	66 (32.5%)	52 (53.1%)	46 (46.9%)	85 (81.0%)	20 (19.0%)	107 (74.4%)	45 (29.6%)	30 (58.8%)	21 (41.2%)

Data are presented as samples of absolute and relative frequencies of each stage of development for each gender and sport.

**Table 3 jfmk-06-00084-t003:** Relative odds of use protective supplies based on the development stage.

						CI (95%)	
Development Stage			χ^2^ (1)	*p*	OR	Lower	Upper
Childhood	vs	Teenagers	5.94	<0.05	3.44	1.21	9.75
		Young adults	29.7	<0.001	10.7	3.99	28.7
		Adults	30.4	<0.001	9.73	3.78	25.0
Teenagers		Young adults	11.6	<0.001	3.12	1.59	6.09
		Adults	11.9	<0.001	2.83	1.54	5.19
Young adults		Adults	-	*Ns*	-	-	-

*Ns* = No significative.

**Table 4 jfmk-06-00084-t004:** Percentage of types of used protection regarding gender and practiced sport.

Gender	Sport	None	Helmet	Spinal Protector	Padded Shorts	Wrist Protector	Knee Protector
**Female**	Ski	12.5	87.5	0.0	0.0	0.0	0.0
	Snow	12.8	74.4	5.1	7.7	0.0	0.0
**Male**	Ski	31.2	68.8	0.0	0.0	0.0	0.0
	Snow	41.7	48.8	3.6	1.2	3.6	1.2

**Table 5 jfmk-06-00084-t005:** Percentage of types of used protection regarding the range of age.

	None	Helmet	Spinal Protector	Padded Shorts	Wrist Protector	Knee Protector
**Childhood**	4.7	95.3	0.0	0.0	0.0	0.0
**Teenagers**	14.5	83.6	0.9	0.0	0.0	0.9
**Young adults**	34.7	61.4	1.0	1.0	2.0	0.0
**Adults**	32.5	64.0	1.5	1.5	0.5	0.0

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
