# Peer review of "Protections in the Recreational Practice of Ski and Snowboard—An Age and Gender Discussion? A Case Study in Spain"

_jfmk, 2021, doi:10.3390/jfmk6040084_

Round 1
Reviewer 1 Report
Dear authors,
the version I received is the same I had previously revised (except for some changes in the references used).
Author Response
Dear reviewer 1,
Thank you for all your effort and valuable recommendations during all the process.
The authors of the present study have changed some parts of the manuscript trying to improve the quality of it.
Reviewer 2 Report
Dear authors,
Thank you very much for the opportunity of reviewing this manuscript. The possible implications of this study are high and novelty, however the procedure for answering to the objectives of the study is incorrect. Firstly, authors must perform a chi-squared test for determining the use (or not) of protective equipment based on gender (male/female), sport modality (shy/snow board) and development (childhood, teenagers, youth and adults) and it could be desirable to include the graphic representation using figures. In the chase of statistical differences, must be included the IC of OR. Then, must be included different tables with the use of different protection based on the total of the sample and different categorization including every statistical analysis (essential the OR with IC). Regarding to the OR interpretation, it necessary to remember that if the lower value of IC is >1 exist a relative risk whereas a lower value of IC <1 exist a protective factor. These analysis are essential because actually the interpretation of OR on the study is wrong and it fault a lot of statistical analysis. It´s necessary to present the statistical treatment proposed because it could affect to the discussion section and conclusions of the study.
While it´s necessary to hope the new statistical treatment for reviewing the results, discussion and conclusions section on the next revision, I would include the next concerns about the introduction and materials and methods sections:
- I don´t know the reason why authors have underlined parts of the manuscript. Has been a mistake?
- It´s necessary to include the code of the Ethics Committee.
- Why authors exclude high-level athletes? There is any relation between the protective equipment – injuries – competitive level. In addition, what was considered high-level competition?
- How was calculated sample size?
- I have a question for authors, how affect the use of protective equipment on the risk of having an injury? This effect of the equipment on the technical-tactical aspects is very important.
Author Response
Reply to Reviewers’ Comments
First of all, we would like to thank you for the time you have spent on your thorough correction of our article. Your comments and suggestions have undoubtedly greatly contributed to its improvement. All changes are marked in greenthroughout the text.
Reviewer comments
Reviewer 2:
Thank you very much for the opportunity of reviewing this manuscript. The possible implications of this study are high and novelty, however the procedure for answering to the objectives of the study is incorrect.
While it´s necessary to hope the new statistical treatment for reviewing the results, discussion and conclusions section on the next revision, I would include the next concerns about the introduction and materials and methods sections, sport modality (shy/snow board) and development (childhood, teenagers, youth and adults) and it could be desirable to include the graphic representation using figures. In the chase of statistical differences, must be included the IC of OR. Then, must be included different tables with the use of different protection based on the total of the sample and different categorization including every statistical analysis (essential the OR with IC). Regarding to the OR interpretation, it necessary to remember that if the lower value of IC is >1 exist a relative risk whereas a lower value of IC <1 exist a protective factor. These analysis are essential because actually the interpretation of OR on the study is wrong and it fault a lot of statistical analysis. It´s necessary to present the statistical treatment proposed because it could affect to the discussion section and conclusions of the study.
Author's reply:
Thank you very much for your valuable comments.
- Analysis of OR have been done to establish odds, and the CI when it has been necessary, between sport modality (shy/snow board) (lines 167-168) and development (childhood, teenagers, youth and adults) (table 3).
- Regarding the graphic representation, we do not know which kind of figure are the correct to express OR. Please, we would be grateful for the reviewer show us with type of graph could fit on our study.
- In relation to the use of different protection, table 5 shows, in a descriptive way, the use of these equipment. However, and from our point of view, due to the distribution of the data (table 5) there is not any sense to display a subsequent statistical analysis of this data.
While it´s necessary to hope the new statistical treatment for reviewing the results, discussion and conclusions section on the next revision, I would include the next concerns about the introduction and materials and methods sections:
I don´t know the reason why authors have underlined parts of the manuscript. Has been a mistake?
Author's reply:
These colored parts have been reviewed in a previous revision.
It´s necessary to include the code of the Ethics Committee.
Author's reply:
We attach, in supplementary files space, the decision of the Ethics Committee about this study. However, there is a lack of code, due to the fact that this Ethics Committee did not use this kind of codes before 2020.
Why authors exclude high-level athletes? There is any relation between the protective equipment – injuries – competitive level. In addition, what was considered high-level competition?
Author's reply:
We decided to exclude high-level athletes because the aim of the study was focused on recreational participants.
How was calculated sample size?
Author's reply:
We used the sample size formula for infinite population formula: n= . The result was n=384,16 and the study had N=520.
I have a question for authors, how affect the use of protective equipment on the risk of having an injury? This effect of the equipment on the technical-tactical aspects is very important.
Author's reply:
Undoubtedly, this question is interesting. However, our study only focused on the use or not of protective equipment. For this reason, the present study cannot answer this valuable question.

Reviewer 3 Report
The aim of this study was to examine the frequency and type of protections used in a sample practicing winter sports in relation to gender and developmental stage.
The topic in interesting and has the merit to provide the reader with additional and extended knowledge on injury risk and correlates.
Here specific comments to the authors:
line 50: please add a comma before "which". This applies throughout the manuscript.
line 58: "from 15 to 60%"
lines 65-67: please rephrase to improve clarity. It comes with a poor English language structure.
lines 77-82: I think this part should be moved to the discussion section or perhaps earlier within the introduction section in the attempt to reinforce the rationale
line 85: "a cross-sectional study design"
line 87-92: what about data coming from under age?
line 99: perhaps it should be appropriate replacing the term "youth" to "young adults" or similar. Of note, the term "youth" embraces both preadolescent and adolescent individuals and used as it is may lead to confusion.
line 122: what about data distribution or normality? please add some specific information
Author Response
Reply to Reviewers’ Comments
First of all, we would like to thank you for the time you have spent on your thorough correction of our article. Your comments and suggestions have undoubtedly greatly contributed to its improvement. All changes are marked in bluethroughout the text.
Reviewer comments
Reviewer 3:
The aim of this study was to examine the frequency and type of protections used in a sample practicing winter sports in relation to gender and developmental stage.
The topic in interesting and has the merit to provide the reader with additional and extended knowledge on injury risk and correlates.
Author's reply:
Thank you very much for your comments.
Here specific comments to the authors:
line 50: please add a comma before "which". This applies throughout the manuscript.
Author's reply:
Thank you very much for your comment. A comma before “wich” was added troughout the manuscript.
line 58: "from 15 to 60%"
Author's reply:
Thank you very much for your comment. “among” was replaced by “from” (L 58)
lines 65-67: please rephrase to improve clarity. It comes with a poor English language structure.
Author's reply:
Thank you very much for your comment. This sentence was rewritten (L 65-67)
lines 77-82: I think this part should be moved to the discussion section or perhaps earlier within the introduction section in the attempt to reinforce the rationale
Author's reply:
Thank you very much for your comment.
line 85: "a cross-sectional study design"
Author's reply:
Thank you very much for your comment. This sentence was modified (L 86)
line 87-92: what about data coming from under age?
Author's reply:
Thank you very much for your comment. A new paragraph was added to explain the underage information: “The privacy of the data given by the subjects was respected and they were also previously informed of the aim, recording techniques and data analysis through informed consents directed to adults and underage” (L 89-90)
line 99: perhaps it should be appropriate replacing the term "youth" to "young adults" or similar. Of note, the term "youth" embraces both preadolescent and adolescent individuals and used as it is may lead to confusion.
Author's reply:
Thank you very much for your comment. “youth” was replaced by “young adults” throughout the manuscript
line 122: what about data distribution or normality? please add some specific information
Author's reply:
Thank you very much for your comment.

Round 2
Reviewer 2 Report
Dear authors,
I want to grant that you have replied and considered point by point to my comments and suggestions. Specially, I would like to grant the inclusion of the IC of OR as the meaning. I believe that the interpretation of the results have improved. Nevertheless, regarding to table 5, I agree with authors. Nevertheless, is it possible that it exist differences in a chi-squared test or relative risk assessed by OR in the use of helmet between sky and snowboard in male or female? I have doubts about it and I suggest to author to perform this analysis for detecting a possible difference and amply descriptive data.
Regarding to the graphic, a diagram with frequencies (expressed as %) is more suitable than figure 2.
Author Response
Reply to Reviewers’ Comments
First of all, we would like to thank you for the time you have spent on your thorough correction of our article. Your comments and suggestions have undoubtedly greatly contributed to its improvement. All changes are marked in greenthroughout the text.
Reviewer comments
Reviewer 2:
I want to grant that you have replied and considered point by point to my comments and suggestions. Specially, I would like to grant the inclusion of the IC of OR as the meaning. I believe that the interpretation of the results have improved. Nevertheless, regarding to table 5, I agree with authors. Nevertheless, is it possible that it exist differences in a chi-squared test or relative risk assessed by OR in the use of helmet between sky and snowboard in male or female? I have doubts about it and I suggest to author to perform this analysis for detecting a possible difference and amply descriptive data.
Regarding to the graphic, a diagram with frequencies (expressed as %) is more suitable than figure 2.
Author's reply:
Thank you very much for your valuable comments.
- New chi-squared test was displayed and the subsequent OR, due to the fact that the results was significant. The OR test showed a higher tendency in skier (OR 3.18) and snowboarder (OR 9.97) women than men in the use of helmet vs no use of any protective equipment than men (Lines 204-210).
- We have removed the figure 1, because it could be difficult to understand. Furthermore, the table 5 provides almost the same information in a clearer way.

This manuscript is a resubmission of an earlier submission. The following is a list of the peer review reports and author responses from that submission.
Round 1
Reviewer 1 Report
Dear authors,
My suggestions about minor correction of your research are in the attachments.
Kind regards

Author Response
We appreciate very much your constructive comments, useful information and your time. Thanks to this review, our manuscript was substantially improved. Responses to your comments are written in bold and highlighted in the manuscript in yellow.
Reviewer #1
General Comments:
The manuscript describes the frequency of the protection used by the skiers and snowboarders at two spanish winter stations. The methodology used was efficient and clear – the use of simple questionnaire. The sample size was high and representative and the study seems well performed. The field of research is extremely important. The results of the study are applicable to skiing safety measures. However, minor revisions are suggested before publishing.
- Thanks to the reviewer 1 for these words. We hope to give new information about this topic.
Table 1: The sum of the last cell should be 520 and not 350 (use of protection: Yes (228) + No (122)…)
- Thanks for this appointment, it was a mistake typing the table, because the percentage was correct. Data were corrected.
Line 154: With the women, it seems (according to table 2) the group that uses the least protection was the adult group (19% vs 81%) not the youth group (16% vs 84%)
- This statement was reviewed and corrected in results and the rest of the manuscript.
Line 164: The word ‘skiers’ should be added. ‘…both men and women skiers use only helmet…’
- This sentence was improved with the suggestion.
Line 165: I do not understand: women use wrist and knee protectors used by men…
- There was a mistake, the sentence was corrected.
Table 4, last two columns: wrist and knee protectors should somewhere have values different that 0,0.
- Thanks for the suggestions. All the results were reviewed and corrected when appropriate.
Line 187: The conclusion that more frequent usage of helmet by the women compared to men is due to fear against the skiing practice is questionable or just partially true. It might as well be due to the fact that women better stick to the rules/recommendations than men or that they want more to be sample to their children…Please try to find some more references on the theme. Moreover, is there any obligation by the law about using the helmet at skiing in Spain? For example, in some countries it is obligatory to use it for some age groups (for instance: children up to 14 years in Slovenia). Please explain the law about helmet use in Spain.
- Thanks for this interesting suggestion. Regarding this, the use of helmet in Spain is not mandatory. For this reason, we did not include more explanation regarding the law regulation.
Line 221: …stablished… should be ‘established ‘
- Word corrected.
Line 229: Grammarly wrong sentence. Please correct.
- This sentence was rewritten.
Line 284: According to the results it should be ‘specially in snowboard’ not ‘specially in
ski’.
- This sentence was corrected.
The interesting finding was that at young ages snowboarders use protections in higher percentages than skiers, however in later ages this changes dramatically. It would be interesting to know whether some participants used to ski or board on jumps or half-pipes or all of them conducted just ‘normal skiing/boarding’. Here I speculate that perhaps teen snowboarders are the ones that execute possibly danger acrobatics on jumps and half-pipes in higher percentage than other ski station participants and this group generally use helmet and possibly other protective equipment as well. In short, the way of skiing/snowboarding (‘normal’ vs jumps/half-pipes users) might have influence on protection use. Please discuss this if you have any data on the way of skiing/boarding at the two stations.
- Thanks for this interesting suggestion and reflection. Unfortunately, we do not have some type of information regarding the way of skiing neither boarding at these two stations. We consider this suggestion for further studies on this research line.

Reviewer 2 Report
The manuscript does not present a relevant study problem to be published in an international journal. The statistical analysis was most descriptive. Given that, the data (and results presented) could be of interest at a national/local level, i.e., authors should try to publish the manuscript in a journal with a more adequate scope, and with a national visibility. Furthermore, the manuscript needs to be increased in some aspects, especially in the introduction and in methods section.
Introduction
- Some references should be checked, since they seem not to be adequate for some ideas presented.
- What is the relevance of the study, and its novelty?
Methods
- Some aspects should be better described. In addition, there are some concerns regarding the instrument used and procedures adopted.
- The age range is too large, and the same questionnaire was used for children aged 6 and also for adults aged 60. Does it make sense?
- The statistical analysis is not clearly presented. For example, authors say “To study the stage of development (childhood, teenagers, youth, and adults) the Pearson χ2” – what does it mean? Moreover, it is also quite descriptive, and more robust tests could be used to better explore the data.
Results
- It is reported “a total of 520 recreational athletes” – does it make sense, call them “athletes”? A 6-year-old child can be called as athlete?
- At the methods section, authors say the sample comprised subjects aged between 6-64 years, but in results they reported a sample aged between 6-50 years. Which one is the correct information?
- Authors presented results related to odds ratio, but it was not presented in the statistical analysis section - how was it computed?
Author Response
We appreciate very much your constructive comments, useful information and your time. Thanks to this review, our manuscript was substantially improved. Responses to your comments are written in bold and highlighted in the manuscript in green.
Reviewer #2
The manuscript does not present a relevant study problem to be published in an international journal. The statistical analysis was most descriptive. Given that, the data (and results presented) could be of interest at a national/local level, i.e., authors should try to publish the manuscript in a journal with a more adequate scope, and with a national visibility. Furthermore, the manuscript needs to be increased in some aspects, especially in the introduction and in methods section.
Introduction
- Some references should be checked, since they seem not to be adequate for some ideas presented.
- All the references of the study have been checked regarding the information linked. After that, some references have been deleted.
- What is the relevance of the study, and its novelty?
- Different studies have studied the injury incidence in winter sports, especially in elite athletes. However, no previous studies have analyzed the type of protections used in these sports, neither have compared the different use regarding gender and development stage. This study can help to adopt new preventative strategies regarding the type of sport, de-fine different messages to increase the frequency of use of protections according to their development stage and, also, to the gender. This appointment has been included in the manuscript.
Methods
- Some aspects should be better described. In addition, there are some concerns regarding the instrument used and procedures adopted.
- Methods subsections have been fully reviewed and details regarding instruments and procedures have been developed to expand the explanation.
- The age range is too large, and the same questionnaire was used for children aged 6 and also for adults aged 60. Does it make sense?
- This manuscript tries to explain and compare the different use of protections regarding the development stage. For this reason, authors consider that the questionnaire was adequate for the whole range of age, similar to previous studies (Fenerty et al., 2013; Ekeland et al., 2018).
- The statistical analysis is not clearly presented. For example, authors say “To study the stage of development (childhood, teenagers, youth, and adults) the Pearson χ2” . What does it mean? Moreover, it is also quite descriptive, and more robust tests could be used to better explore the data.
- Thanks for the suggestion. New and more robust analyses have been performed to explore better our database.
Results
- It is reported “a total of 520 recreational athletes” – does it make sense, call them “athletes”? A 6-year-old child can be called as athlete?
- The term has been changed to avoid misconceptions.
- At the methods section, authors say the sample comprised subjects aged between 6-64 years, but in results they reported a sample aged between 6-50 years. Which one is the correct information?
- The manuscript has been fully reviewed to correct the data. The correct range is 6-64 years.
- Authors presented results related to odds ratio, but it was not presented in the statistical analysis section - how was it computed?
- This information has been included in the statistical analysis.
Round 2
Reviewer 2 Report
Despite authors tried to improve the manuscript, it does not look to be enough to be published in the journal. The previous suggestion (to publish in a national/local journal) looks to be the more adequate (but changes and quality improvement are still required). Furthermore, some points still need for attention. Such as:
- Some references do not look to be adequate for the idea presented. For example, see references number 1 and 2, in the introduction – do they really present the information cited in paragraph one?
- Authors included information related to the pilot study. What was the age range of the sample?
- Once more, statistical analysis was not clearly presented. For example: What was the statistical procedure used to estimate the OR? More, authors pointed “To study the stage of development (childhood, teenagers, youth, and adults) the Pearson c2 was used along with the contingency coefficient”, but authors did not study “the stage of development”.
- In methods section, authors say that subjects were split into categories related to “development stages”, but in line 138 they say that sample was stratified according to “educative stages”. Are they both the same?
- Figure 1 is not easy to understand, and its quality should be improved.